



1    Using the anomaly forcing Community Land Model (CLM 4.5) for crop yield projections

Yaqiong Lu[1,2*] and Xianyu Yang[3]
[1]Institute of Mountain Hazards and Environment, Chinese Academy of Sciences, Chengdu
610040, China
[2]National Center for Atmospheric Research, Boulder, CO 80305, USA
[3]Chengdu University of Information Technology, Chengdu, 610225, China
*Corresponding author: Yaqiong Lu, yaqiong@imde.ac.cn, 8602861158015, #.9, Block 4 ,
Renminnanlu Road, Chengdu 610041, China

Abstract
Crop growth in land surface models normally requires high temporal resolution climate data (3-
hourly or 6-hourly), but such high temporal resolution climate data are not provided by many
climate model simulations due to expensive storage, which limits modeling choice if there is an
interest in a particular climate simulation that only saved monthly outputs. The Community Land
Surface Model (CLM) has proposed an alternative approach for utilizing monthly climate outputs
as forcing data since version 4.5, and it is called the anomaly forcing CLM. However, such an
approach has never been validated for crop yield projections. In our work, we created anomaly
forcing datasets for three climate scenarios (1.5 °C warming, 2.0 °C warming, and RCP4.5) and
validated crop yields against the standard CLM forcing with the same climate scenarios using 3-
hourly data. We found that the anomaly forcing CLM could not produce crop yields identical to
the standard CLM due to the different submonthly variations, and crop yields were underestimated
by 5-8% across the three scenarios (1.5 °C, 2.0 °C, and RCP4.5) for the global average, and 28-
41% of cropland showed significantly different yields. However, the anomaly forcing CLM
effectively captured the relative changes between scenarios and over time, as well as regional crop
yield variations. We recommend that such an approach be used for qualitative analysis of crop
yields when only monthly outputs are available. Our approach can be adopted by other land surface
models to expand their capabilities for utilizing monthly climate data.

Key words: Community Land Model; Crop yields; Anomaly forcing


Introduction

Increasing numbers of future climate scenarios exhibit large uncertainties for crop yield projections.
Crop yields may increase or decrease depending on which climate projection is used (Lobell et al.
2008, Urban et al. 2012, Rosenzweig et al. 2014). Ensemble future climate projections, such as
CMIP5, showed a large range of future climate projections, even for one emission scenario (Knutti
and Sedlacek 2013). Using all future climate projections is not realistic not only because of the
computational expense but also because many of these future climate projections only save
monthly climate outputs that are not suitable for crop models that require high temporal resolution
forcing data. Some standalone process-based crop models run in daily time steps, and some crop
models embedded in land surface models need at least 6-hour climate data as the forcing data to
represent diurnal cycles. Only a small portion of the CMIP5 (Coupled Model Intercomparison
Project 5) simulations (<25%) can be used as the forcing data for crop models, leaving little room


for crop modelers to choose a particular climate model projection that is of interest.

The Community Land Model (CLM) (Oleson et al. 2013) is a state-of-the-art land surface model
that simulates biogeophysical (radiation transfer, vegetation-soil-hydrology, surface energy fluxes,
etc.) and biogeochemical (soil carbon and nitrogen cycle, vegetation photosynthesis, dynamic
vegetation growth, etc.) processes. CLM is the default land model in the Community Earth System
Model (CESM) (Hurrell et al. 2013), and it can be run either online coupled with the rest of CESM
(atmosphere and ocean) or offline (the land model only, forced with climate datasets) for multiple
spatial extents (site, regional, and global) and at different resolutions. The crop model derived from
AgroIBIS (Kucharik 2003) was introduced to CLM4.0 by Levis et al. (2012), and it is responsible
for crop growth phenology (temperature determined), carbon allocation algorithms, and crop
management (e.g., irrigation). The crop model in CLM runs when the biogeochemical compset is
active, and it was tested with the CLM-CN compset in version 4.0 and tested with CLM-BGC in
version 4.5. Since their introduction, crop models in the CLM have been developed to represent
more crop types and processes, such as soybean nitrogen fixation (Drewniak et al. 2013), ozone
impacts on yields (Lombardozzi et al. 2015), winter wheat growth responses to cold hazards (Lu
et al. 2017), and maize growth responses to heat stress (Peng et al. 2018). CLM simulates nine
crop types, accounting for 54% of global total crop production (other production is represented by
the most similar crop type): maize, soybean, spring wheat, winter wheat, cotton, rice, sugarcane,
tropical maize, and tropical soybean. In this study, we used CLM version 4.5 (Oleson et al. 2013).

Since version 4.5, CLM offers a built-in function that indirectly uses monthly climate outputs as
the forcing data, and is called the anomaly forcing CLM (Lawrence et al. 2015). Anomaly forcing
CLM reconstructs new subdaily forcing data by applying the precalculated future monthly
anomaly signals to user-defined historical subdaily forcing data, referred to as the reference data.
The future monthly anomaly signals are calculated by the future monthly climate outputs and by
use of historical monthly outputs. The choice of reference data is arbitrary. Any existing subdaily
forcing data (e.g., CRUNCEP, QIAN) for CLM can be used as the reference data. The historical
monthly outputs are recommended to be multiyear averaged to represent the historical means and
avoid affecting the monthly anomaly signal by rare, extreme events in a particular year. Such an
arbitrary choice is because the goal of the original anomaly forcing CLM is not to reconstruct
future forcing that is identical to the actual future forcing when the high temporal resolution data
were saved. Rather, the original goal of the anomaly forcing CLM is to understand the influences
due to the anomaly signal by comparing the simulation with the anomaly forcing CLM to the
simulation run with the reference data. The differences between the two simulations are due to the
anomaly signals.

In our study, we modified the anomaly forcing CLM to fit our goals to understand whether we
could simply use the anomaly forcing CLM for crop yield projections when only monthly climate
data were available. We carefully chose the historical monthly data and the reference data so that
the reconstructed future anomaly forcing had nearly identical monthly means as the desired
subdaily future forcing, but we used different submonthly variations. We created anomaly forcing
datasets for three future scenarios (1.5 °C warming, 2.0 °C warming, and RCP4.5) for 2006-2075
for which both the subdaily and monthly climate outputs were available from three CESM
simulations. With the three paired CLM simulations, we validated the anomaly forcing CLM by
comparing it to the standard CLM.




Methods

The original anomaly forcing CLM has been in function since CLM4.5. This approach reconstructs
the subdaily (3-hourly or 6-hourly) forcing data by applying the monthly anomaly signal to user
selected subdaily reference data; therefore, it indirectly uses the monthly atmospheric outputs as
the forcing data for CLM. This approach does not change any of the scientific code in CLM; it
only adds code that reads the monthly anomaly signals and automatically applies these to the
reference data while the CLM is running. There were two monthly anomaly signals for RCP4.5
and RCP8.5 that were generated using the CESM future projections and were ready for use. It is
the user's choice to select which subdaily reference (e.g., CRUNCEP or CLMQIAN) and which
years to use. By simply modifying user_nl_cpl namelist and adding data streams of the anomaly
forcing variables (see the appendix for the detailed usage), the anomaly forcing CLM will
automatically read the monthly anomaly signal and apply the signal to each time step of the
reference data within a month. When the reference data period is less than the anomaly signal
period, the anomaly forcing CLM will cycle the same reference data until the simulation is
complete. Because the different selections of reference data can generate different forcings, even
with the same monthly anomaly signals, one should not use the simulation from the anomaly
forcing CLM to represent the actual simulation. Rather, the original goal of the anomaly forcing
CLM is to compare the simulation with the anomaly forcing and simulation with the reference
forcing data to understand the effects of the monthly anomaly signals on land surface variables.

The goal of this work is to test how well crop yield projections from the anomaly forcing CLM
compare to the projections from the standard forcing CLM, given that anomaly forcing has the
same monthly average as standard forcing. We selected three future scenarios for CESM
simulations that saved both monthly outputs and 3-hourly outputs, where the 3-hourly outputs
were directly used in the standard forcing CLM, and the monthly outputs were indirectly used in
the anomaly forcing CLM. We calculated the anomaly forcing signals using the monthly CESM
outputs and the monthly average of reference data, so that when applying the anomaly signals to
the reference data, it is expected to generate identical monthly means as does regular forcing.
However, due to a limit in calculations of precipitation anomalies (precipitation anomaly ratio less
than 5 times) and how the CLM treats snow and rainfall, the anomaly forcing CLM did not show
identical snow and rainfall monthly averages and introduced bias in the crop yield simulations (see
the results section).

Table 1. A summary of the original anomaly forcing CLM and the modifications in this work

|  | Original anomaly forcing CLM | Modifications in this work |
|---|---|---|
| 3 h/6 h reference data | User choice | 6 h CAM outputs from one historical low warming ensemble simulation 1996-2005 |
| Monthly anomaly signals | Existing for RCP4.5 and RCP8.5 | • Anomalies between future scenarios and monthly means of reference data |

|  |  | • Three future scenarios: 1.5 °C, 2.0 °C, and RCP4.5<br>• Each scenario had monthly outputs and 3 h outputs |
|---|---|---|
| Goals | Climate impact due to anomaly signals when comparing the anomaly run with the reference run | Given that anomaly forcing has the same monthly mean as MOAR, can we use it for crop yield projections? |

We randomly chose the 6-hourly reference data (1996-2005) from one of the 11 historical low
warming ensemble CESM simulations. Additionally, we selected three CESM future simulations
for the 1.5 °C warming, 2.0 °C warming, and RCP4.5 scenarios, where all the three simulations
saved both the monthly outputs and the 3-hourly outputs. We then calculated the monthly anomaly
signal at each grid cell for each scenario (1.5, 2.0, and RCP45) from 2006-2075. The monthly
anomaly signals are differences for temperature, specific humidity, wind, and air pressure and are
ratios for solar radiation and precipitation between the monthly outputs of each scenario and the
1996-2005 averaged monthly values of the reference data. The anomaly forcing signal has both
spatial and monthly variations. When running the anomaly forcing simulation for 2006-2070,
CLM repeatedly uses the 10-year reference period and applies the anomaly signal of a month to
all subdaily reference forcing in this month. For example, an anomaly forcing simulation for 2006
January uses the 1996 January reference data plus or multiplies (if the anomaly signal is a ratio)
the 2006 January anomaly signal. If the 2006 January temperature anomaly is 1 K for a grid cell,
then all 1996 January reference data will be increased by 1 K for the grid cell.
The monthly anomaly signal is calculated at each grid cell (i,j). For temperature, pressure, wind,
and humidity, the anomaly signal is the difference between the future monthly data and the
historical monthly average (equation 1). For solar radiation, longwave radiation, and
precipitation, the anomaly signal is the ratio between the future monthly data and the historical
monthly average (equation2). We set the maximum ratio for precipitation to 5 to avoid unrealistic
extreme precipitation, which also introduced biases in precipitation (discussed in the discussion
section).
$$var\_af_{i,j,m} = fut\_var_{i,j,m} - \overline{hist\_var_{i,j,m}} \qquad (1)$$
$$var\_af_{i,j,m} = fut\_var_{i,j,m} / \overline{hist\_var_{i,j,m}} \qquad (2)$$
We set up global CLM crop simulations (compset CLM45BGCCROP) at 1.9 by 2.5 in latitude and
longitude, respectively, using the anomaly forcing CLM and the regular forcing CLM for the 1.5 °C
warming, 2.0 °C warming, and RCP4.5 scenarios. All simulations used the default nitrogen
fertilization rates and a constant $CO_2$ level of 359.8 ppm. For each scenario, we validate the crop
yield in the anomaly forcing CLM to the regular forcing CLM to determine if we can use the



anomaly forcing CLM for future crop yield projections. We also studied whether the anomaly
forcing CLM has a similar crop growth response to transient CO2 and nitrogen fertilization.
However, due to limited computational resources, we only tested such responses for the RCP4.5
scenario. The transient CO2 levels in the RCP45 scenario gradually increased from 379 ppm in
2006 to 530 ppm in 2070. To test the nitrogen fertilization effects, we simply added a zero nitrogen
fertilization simulation here.
We adopted the two-sample Kolmogorov-Smirnov test (KS test) to test the statistical significance
of differences between the anomaly forcing CLM and the standard CLM for atmospheric forcing
data and yield. We used the KS test because some variables at some grid cells did not necessarily
follow normal distributions. The KS test is a nonparametric test that detects differences in the
empirical probability distributions between two samples, and the two samples do not need to have
normal distributions (Justel et al. 1997, Marozzi 2013). When repeated using the ten-year reference
data, we expected that the ten year averaged monthly anomaly forcing would show no significant
differences from the regular forcing. Thus, for the atmospheric forcing data, we tested probability
distribution differences between anomaly forcing and regular forcing for every ten-year averaged
monthly dataset (sample size was 7x12=84). For crop yields, we used the every ten-year averaged
annual yields (sample size was 7). We used linear regression coefficent ($R2$), bias (equation 3),
percentage differences (equation 4) in our evaluations.

$$bias = CLM_{anomaly\ forcing} - CLM_{standard} \quad (3)$$

$$\%differences = 100 * (\frac{CLM_{anomaly\ forcing}}{CLM_{standard}} - 1) \quad (4)$$

Results
We aimed to generate an anomaly forcing that produced identical monthly averages as its
counterpart regular forcing (the desirable 3-hourly forcing data for CLM) but with different
submonthly variations. All atmospheric forcing variables achieved this goal except for
precipitation and its liquid and ice components, rain and snow. The linear regression coefficients
($R^2$) between anomaly forcing and standard forcing for the monthly means of incoming solar
radiation, bottom atmosphere temperatures, pressures, humidities, and winds all showed $R^2$ values
above 0.99, and there were also no significant differences for these variables for all grid cells.
However, for rain and snow, the $R^2$ values were 0.63-0.87 and 0.88-0.96 across the three scenarios,
respectively (Figure 1a). Statistically significant differences were also found for rain and snow in
many regions in the Northern Hemisphere (Figure 2). We used monthly variances as a measure of
the submonthly variations. $R^2$ for variances of forcing were low for most variables except for
incoming solar radiation (Figure 1b). Such lower $R^2$ values indicated that anomaly forcing could
not represent the submonthly variations as well as the regular forcing.
There were two error sources for precipitation. First, there was overall average lower precipitation
in the anomaly forcing by 0.02 mm/day, 0.03 mm/day, and 0.2 mm/day in the 1.5 °C, 2.0 °C, and
RCP45 scenarios, respectively. Such slightly lower precipitation was because we set the maximum
precipitation anomaly ratio to 5 to avoid unrealistically extreme precipitation levels. Second, the
CLM used the temperature in each time step to determine if the given precipitation was rain or


snow. Precipitation was rain when temperature was above 273.15 K, otherwise it was
snow. Therefore, the different submonthly variations in temperature resulted in different submonthly
variations for snow and rain. Due to this problem, the lower precipitation did not evenly distribute
to the rain and snow bias, for which rain was underestimated by 0.08-0.3 mm/day, and snow was
overestimated by 0.06-0.11 mm/day across the three scenarios. The significantly different regions
were mainly in the Northern Hemisphere and the Antarctic, and most regions in the Southern
Hemisphere did not show significant differences in rain or snow. How the rain and snow biases
affected yield projections will be discussed.

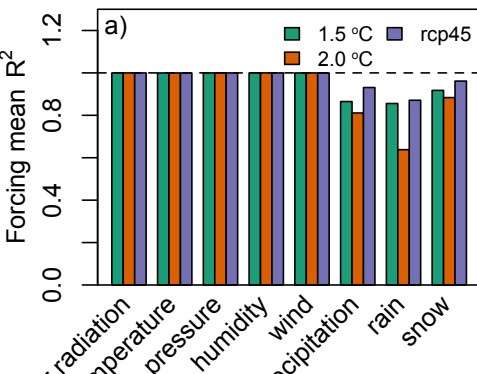 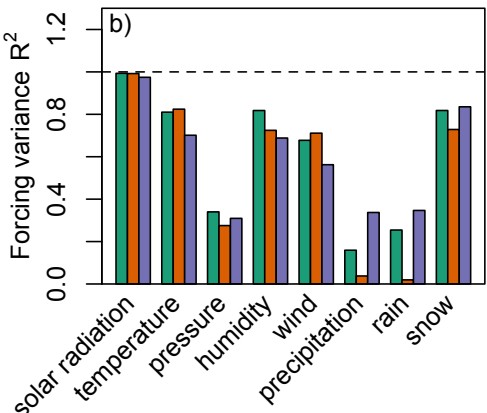

Figure 1. Linear regression coefficients ($R^2$) between a) decade-averaged monthly mean (sample
size =12 months x 7 decades=84) between anomaly forcing and regular forcing and b) every ten
220          year-averaged monthly variance between anomaly forcing and regular forcing.

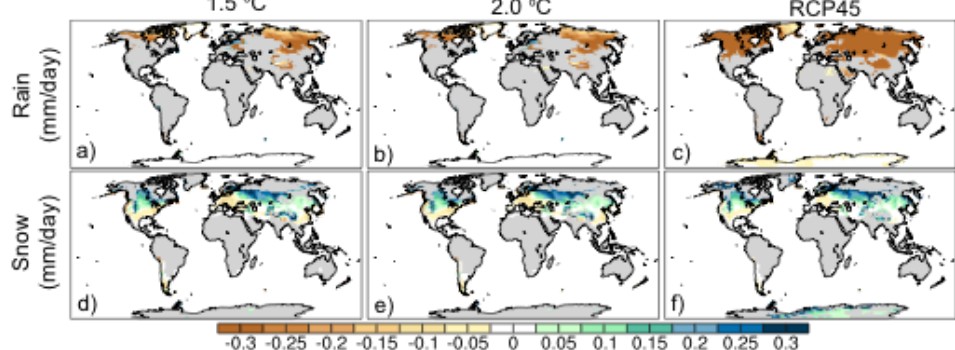

Figure 2. 70-year averaged differences between anomaly forcing and regular forcing for rain (a-





c) and snow (d-f) for the 1pt5, 2pt0, and RCP45 scenarios. All differences shown here are
statistically significant differences tested by the Kolmogorov-Smirnov test with a sample size of
84. The gray areas are regions that did not show significant differences.
When compared to crop yield simulations in the standard CLM, the anomaly forcing CLM
underestimated crop yields by 5-8% across the three scenarios for the global average, and 28-41%
of cropland showed statistically significant differences in yields. The rainfed crop yield differences
across the three scenarios showed largely similar spatial distributions: overestimation in the
northern US and Europe and underestimation in the Southern Hemisphere and in East Asia (Figure
3d-f). The overestimated rainfed crop yield (mainly for maize and wheat) in the anomaly forcing
CLM is due to higher water availability in these regions, which is a result of higher snow in the
anomaly forcing CLM. For irrigated crops, such overestimations in the northern US and Europe
disappear (Figure 3g-i) because sufficient irrigation was added to the irrigated soil column; as long
as there is plant water stress which removed water availability impacts on crop yields. However,
the underestimations in the Southern Hemisphere and East Asia were persistent, because water
availability does not cause yield differences for irrigated crops; we suspect such underestimations
were caused by the other error in forcing data: the different submonthly variations in the forcing
data.

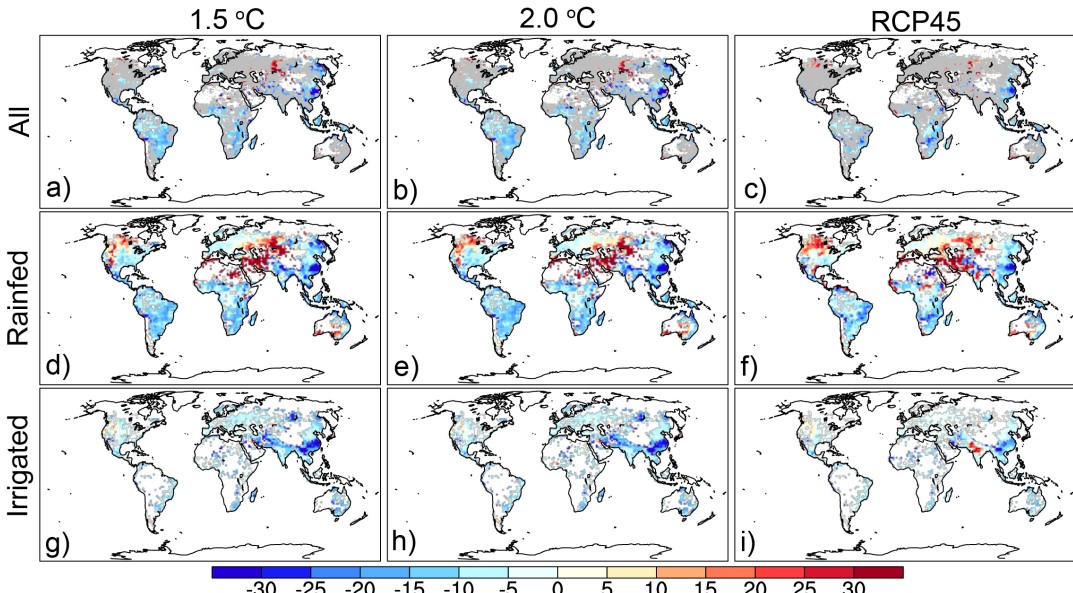

Figure 3. The percentage differences of 70-year integrated yields between the anomaly forcing
CLM and the standard CLM for all crops (a-c), rainfed crops (d-f), and irrigated crops (g-i) for
the 1.5 °C, 2.0 °C, and RCP45 scenarios. The white regions are where no crops grow based on
the historical crop map in 2005. For plots a-c, we showed only the significant differences as
determined by the by Kolmogorov-Smirnov test with a sample size of 7. The regions with
insignificant differences are masked as gray in a-c. For plots d-i, we did not mask the


insignificant differences

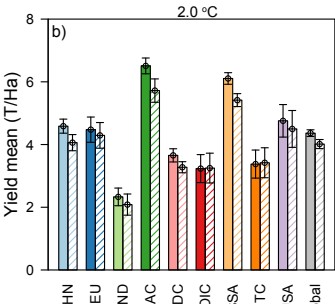
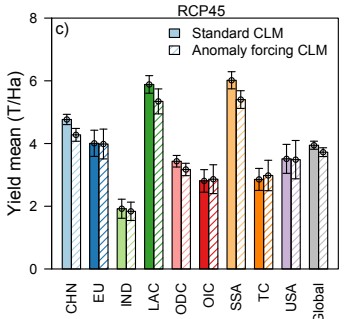

Figure 4. Regional comparisons of the 70-year integrated mean yields and yield standard deviations between the anomaly forcing CLM and the standard CLM. The error bars indicate 70-year yield standard deviations. CHN: China; EU: European Union; IND: India; LAC: Latin America; ODC: Other Developing Countries; OIC: Other Industrialized Countries; SSA: Sub-Saharan Africa; TC: Transition Countries; USA: United States

The global 70-year averaged yields ± standard deviation in the standard CLM and in the anomaly forcing CLM are 4.38 ± 0.09 and 4.03 ± 0.16 t/ha, respectively, in the 1.5 °C scenario, 4.36 ± 0.11 and 4.01 ± 0.14 t/ha, respectively, in the 2.0 °C scenario, 3.95 ±0.13 and 3.72 ± 0.14, respectively, in the RCP45 scenario. The anomaly forcing CLM captured the regional yield variations. Latin America (LAC) showed the highest yield while India (IND) showed the lowest yields for both the anomaly forcing CLM and the standard CLM across the three scenarios.

Although the crop yields were underestimated, the anomaly forcing CLM could qualitatively represent the spatial yield differences between two climate scenarios. Comparing 2.0 °C to 1.5 °C, there was a 4-8% yield increase in the northern U.S. and a 0-4% yield decrease in (Figure 5a) in the southeast U.S. When comparing the RCP45 to the 1.5 °C scenario, crop yields in the U.S. were largely reduced (up to 50%). The anomaly forcing CLM clearly captured these yield differences (Figure 5b and 5d).



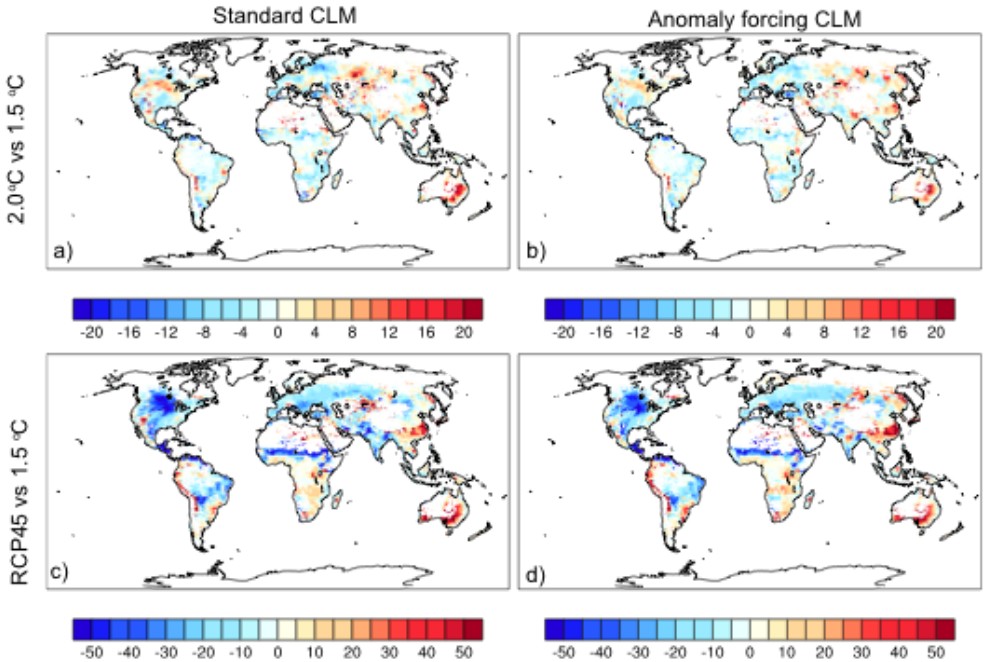

Figure 5. The percentage of 70-year integrated yield differences between 2.0 °C and 1.5 °C (top
panel) RCP45 to 1.5 (bottom panel) in the standard CLM and the anomaly forcing CLM



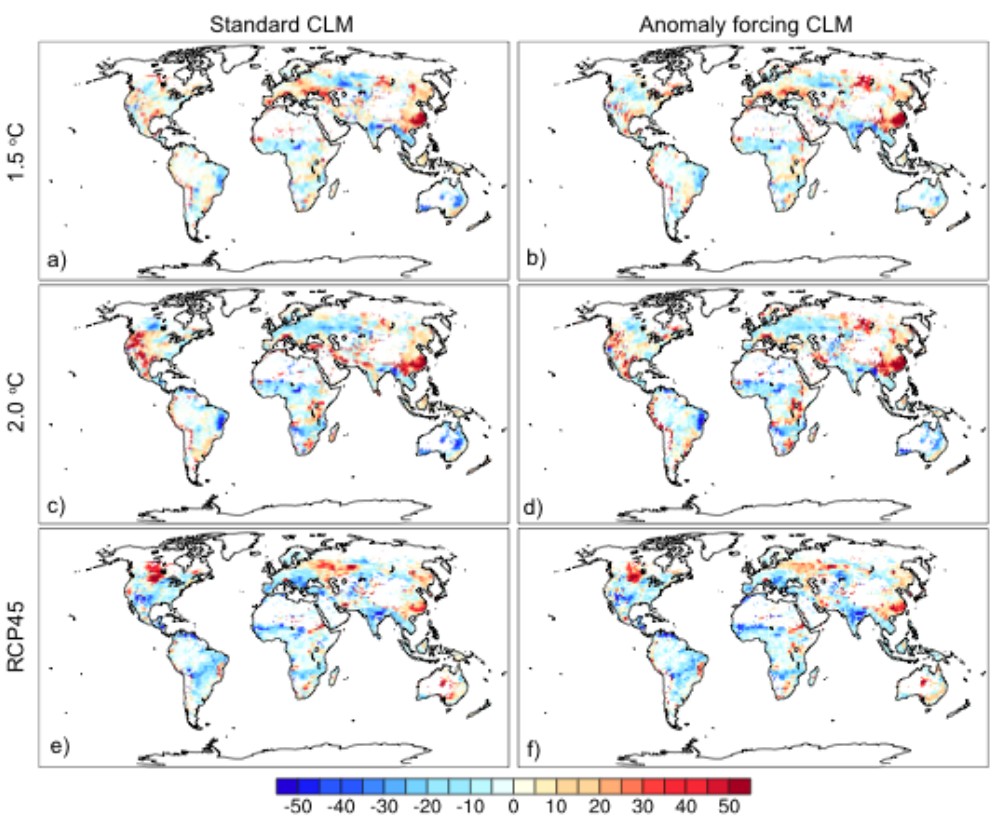

Figure 6. The percentage yield difference from 2006-2015 to 2066-2075 in the standard CLM
and anomaly forcing CLM across the three scenarios

The anomaly forcing CLM also captured yield changes over time for each climate scenario. The three scenarios showed some similarities in yield changes from 2006-2015 to 2066-2075. For example, crop yields increased in Southeast China and decreased in Sub-Saharan Africa. There were also yield changes that were unique to each scenario that were also found in the anomaly forcing CLM. For example, crop yields increased in Europe for the 1.5 °C scenario (Figure 6a-b), while they decreased in Europe for the 2.0 °C and RCP45 scenarios (Figure 6c-f), and crop yields declined in the U.S. for the RCP45 scenario (Figure 6e-f) while they increased for the 1.5 °C and 2.0 °C scenarios (Figure 6 a-d).

All simulations in the above evaluations adopted a constant $CO_2$ level (359.8 ppm) and crop types dependent fixed nitrogen fertilization (25-500 kg N/ ha), so whether the anomaly forcing CLM simulated a similar or different crop growth response to $CO_2$ or nitrogen fertilization is unknown. Due to limited computational resources, we tested crop responses to transient $CO_2$ and nitrogen fertilization only for the RCP45 scenario and assumed that the other scenarios would show the same differences as the RCP45 scenario. The transient $CO_2$ in the RCP45 scenario gradually increased from 379 ppm in 2006 to 530 ppm in 2075. To test the effects of nitrogen fertilization,



we simply added a zero nitrogen fertilization simulation. Although all grid cells had the same
amounts of $CO_2$ increase in a given year (no spatial variation), crop yields had spatial variations
in response to transient $CO_2$. Most regions showed a 5-10% yield increase, but some regions
showed much higher yield increases, such as northern India, the southern edge of the Sahara, and
Australia (Figure 7a). Such crop yield responses to transient $CO_2$ spatial patterns were also
captured by the anomaly forcing CLM (Figure 7b). Similar for the crop yield responses to nitrogen
fertilization, the anomaly forcing CLM simulated crop yield increase spatial patterns (Figure 7c-
d), in which the Southern Hemisphere and Asia had greater yield increases in response to nitrogen
fertilization.

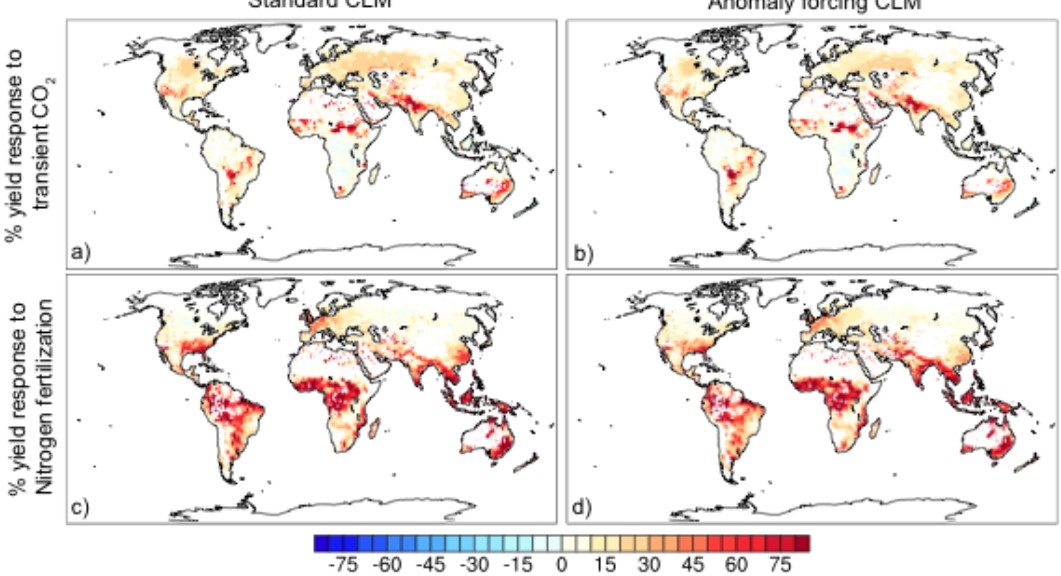

Figure 7. 70-year averaged integrated crop yield response to transient $CO_2$ and to no nitrogen
fertilization in the anomaly forcing CLM (a and b) and in the standard CLM (c and d) for the
RCP45 scenario.

Discussion

In this work, we created anomaly forcing datasets for three future climate scenarios, and we
validated the crop yields in the anomaly forcing CLM by comparison with the crop yields in the
standard CLM. The differences between the anomaly forcing CLM and standard CLM were due
only to differences in forcing data, for which the standard CLM used regular forcing (three-hourly
forcing) and the anomaly forcing CLM used anomaly forcing. We found that the anomaly forcing
CLM underestimated crop yields but identified the regional yield variations, as well as yield
differences between two climate scenarios and yield changes over time. The anomaly forcing CLM
could not generate the exact same crop yields as the standard CLM due to errors in precipitation
and in the submonthly variations. However, it could be used for qualitative analysis of relative



crop yield changes among different scenarios and over time.

The overall underestimation of crop yields may be due to differences in phenology that resulted from different submonthly variations. Some of the low yields in the anomaly forcing CLM may be explained by shorter grain fill periods. For example, the lower rice yields in southeast China is due are due to a 5-10 day shorter grain fill period in the anomaly forcing CLM; maize and soybean in the Southern Hemisphere also showed a 1-5 day shorter grain fill period that may account for the lower yields. In addition to the low yields, the anomaly forcing CLM also simulated lower GPP and LAI compared to the standard CLM, and the spatial distributions of GPP and LAI differences were very similar to the yield differences.

Some regions in the Northern Hemisphere showed higher rainfed crop yields in the anomaly forcing CLM, which is due to higher soil moistures at planting that resulted from higher snow levels in the Northern Hemisphere. Crop growth in CLM is very sensitive to the soil moisture at planting, and higher soil moisture results in unstressed crop growth and hence produces higher yields. When adequate irrigation is applied, both the anomaly forcing and the standard CLM models have sufficient water for crop growth, and the overestimations disappeared. Therefore, the anomaly forcing may not be appropriate for estimating the actual future irrigation demands but is able to distinguish the relative differences in irrigation demand across different climate scenarios.

The energy fluxes in the anomaly forcing CLM and in the standard CLM were different due to different crop growth rates and differences in forcing data. The higher snow cover in the Northern Hemisphere creates higher albedo and lowers absorbed solar radiation and hence lower surface energy fluxes. The higher LAI increased the summer latent heat flux up to 5 $W.m^{-2}$ (not shown), while the annual latent heat flux showed 5-10 $W.m^{-2}$ lower values in the anomaly forcing CLM due to the lower net radiation. In the Southern Hemisphere, lower LAI resulted in lower latent heat fluxes and higher sensible heat fluxes.

The regional yield comparisons indicate that the anomaly forcing CLM effectively captured regional yield variations but with slightly lower yield biases. We want to point out that the very high crop yields in Latin America and in Sub-Saharan Africa, and the very low crop yields in India in both the anomaly forcing CLM and the standard CLM approaches are not realistic when compared to the UNFAO yields. Such biases in the CLM have been discussed by Levis et al. (2018), and the low yields in India are due to incorrect crop phenology when crops entered the grain fill during the dry season. The high yields in Latin American and in Sub-Saharan Africa were due to the nitrogen fertilization amounts based on US levels, which are too high for these regions.

Conclusions

The Community Land Surface model offers an alternative way in utilize the monthly climate as the forcing data. Such an approach could expand user choice of forcing data when high temporal resolution climate data are not available. In this work, we created anomaly forcing data for three climate scenarios (1.5 °C warming, 2.0 °C warming, and RCP4.5) and validated crop yield projections in the anomaly forcing CLM against the standard CLM. The anomaly forcing CLM underestimated crop yields by 5-8%, which was largely due to the differences in phenology and photosynthesis that resulted from the different submonthly variations. How CLM treated



precipitation as rain or snow also introduced biases in crop yields and in the energy flux
simulations. Although the anomaly forcing CLM could not generate crop yields identical to the
standard CLM, it could be used for qualitative analysis of crop yield changes across various
scenarios over time.
Code availability
The CLM source code used in our study is available at repository website Zenodo:
https://doi.org/10.5281/zenodo.3900671
Author contribution
Yaqiong Lu designed and performed the simulations. Yaqiong Lu and Xianyu Yang analyzed the
results and wrote the manuscript.
Acknowledgments

We thank Sean Swenson and David Lawrence for instruction of using the anomaly forcing
approach. This work was supported by the National Science Foundation under Grant Number
AGS-1243095 and the National Natural Science Foundation of China (No. 41975135). We would
like    to    acknowledge    high-performance    computing    support    from    Yellowstone
(ark:/85065/d7wd3xhc), provided by NCAR's Computational and Information Systems
Laboratory, sponsored by the National Science Foundation.
Appendix: a user guide for using anomaly forcing CLM
Running the anomaly forcing CLM is similar to the standard CLM but with several additional
steps. First, the monthly anomaly data are prepared as described in the method section. Then, the
user needs to modify user_nl_cpl and user_nl_datm to specify which forcing variables to add to
the anomaly signals. There are seven anomaly forcing variables (Table A2), and the user can
specify one, or two, or all variables in the two namelists (user_nl_cpl and user_nl_datm). The
final step is to add the corresponding anomaly forcing data streams depending on which anomaly
forcing variables were specified in user_nl_cpl and user_nl_datm.
1.  Modify user_nl_cpl and user_nl_datm
The user may add part or all of the following text to user_nl_cpl.
cplflds_custom        =        'Sa_prec_af->a2x',        'Sa_prec_af->x2l','Sa_tbot_af->a2x',
'Sa_tbot_af->x2l','Sa_pbot_af->a2x',                    'Sa_pbot_af->x2l','Sa_shum_af->a2x',
'Sa_shum_af->x2l','Sa_u_af->a2x',                        'Sa_u_af->x2l','Sa_v_af->a2x',
'Sa_v_af->x2l','Sa_swdn_af->a2x', 'Sa_swdn_af->x2l','Sa_lwdn_af->a2x', 'Sa_lwdn_af->x2l'
Add part or all of the following text into user_nl_datm:





anomaly_forcing=
'Anomaly.Forcing.Precip','Anomaly.Forcing.Temperature','Anomaly.Forcing.Pressure','Anomaly.
Forcing.Humidity','Anomaly.Forcing.Uwind','Anomaly.Forcing.Vwind','Anomaly.Forcing.Short
wave','Anomaly.Forcing.Longwave'
Also attach the anomaly forcing data streams in user_nl_datm:
streams       =       "datm.streams.txt.CLMCRUNCEP.Solar       1996       1996       2005",
"datm.streams.txt.CLMCRUNCEP.Precip 1996 1996 2005",
"datm.streams.txt.CLMCRUNCEP.TPQW           1996           1996           2005",
"datm.streams.txt.presaero.clim_2000 1 1 1",
"datm.streams.txt.Anomaly.Forcing.Precip           2006           2006           2075",
"datm.streams.txt.Anomaly.Forcing.Temperature 2006 2006 2075",
"datm.streams.txt.Anomaly.Forcing.Pressure           2006           2006           2075",
"datm.streams.txt.Anomaly.Forcing.Humidity 2006 2006 2075",
"datm.streams.txt.Anomaly.Forcing.Uwind           2006           2006           2075",
"datm.streams.txt.Anomaly.Forcing.Vwind 2006 2006 2075",
"datm.streams.txt.Anomaly.Forcing.Shortwave           2006           2006           2075",
"datm.streams.txt.Anomaly.Forcing.Longwave 2006 2006 2075",
"/glade/p/work/yaqiong/inputdata/atm/datm7/co2.1pt5degC.streams.txt 1901 1901 2075"
mapalgo = 'bilinear', 'bilinear', 'bilinear', 'bilinear', 'bilinear', 'bilinear', 'bilinear', 'bilinear', 'bilinear',
'bilinear', 'bilinear', 'bilinear','nn'
tintalgo = 'coszen', 'nearest', 'linear', 'linear', 'nearest', 'nearest', 'nearest', 'nearest', 'nearest', 'nearest',
'nearest', 'nearest','linear'
Any combination or subset of anomaly forcing variables can be used. For example,
cplflds_custom = 'Sa_prec_af->a2x', 'Sa_prec_af->x2l' (in user_nl_cpl)
anomaly_forcing='Anomaly.Forcing.Precip' (in user_nl_datm)
will only adjust precipitation. The reference data and period are defined in env_run.xml.
2.  Add the anomaly forcing data stream
The anomaly forcing data stream is where to specify the data path of the monthly anomaly forcing
signal and to tell the code which variable to retrieve. A list of all anomaly forcing data stream file
names and the variables in the anomaly forcing data and the code are given in Table 2. An example
of the content in user_datm.streams.txt.Anomaly.Forcing.Humidity is also attached. The user only
needs to add the corresponding variable data streams that are defined in user_nl_cpl.
Table A2. A list of the anomaly forcing data streams and the corresponding variables in the
anomaly forcing data and the code

| Data stream file names | Vars in data | Vars in code |
|---|---|---|
| user_datm.streams.txt.Anomaly.Forcing.Humidity[1] | huss | shum_af |
| user_datm.streams.txt.Anomaly.Forcing.Precip | pr | prec_af |
| user_datm.streams.txt.Anomaly.Forcing.Pressure | ps | pbot_af |
| user_datm.streams.txt.Anomaly.Forcing.Shortwave | rsds | swdn_af |





| user_datm.streams.txt.Anomaly.Forcing.Temperature | tas | tbot_af |
|---|---|---|
| user_datm.streams.txt.Anomaly.Forcing.Uwind | uas | u_af |
| user_datm.streams.txt.Anomaly.Forcing.Vwind | vas | v_af |
| user_datm.streams.txt.Anomaly.Forcing.Longwave | rlds | lwdn_af |

[1]An example of the content in the data stream was given below:

```
<dataSource>
GENERIC
</dataSource>
<domainInfo>
<variableNames>
time
xc      lon
yc      lat
area
mask
</variableNames>
<filePath>
482         /glade/p/cesmdata/cseg/inputdata/share/domains
</filePath>
<fileNames>
domain.lnd.fv0.9x1.25_gx1v6.090309.nc
</fileNames>
</domainInfo>
<fieldInfo>
<variableNames>
huss  shum_af
</variableNames>
<filePath>
THE ANOMALY FORCING SIGNAL DATA PATH
</filePath>
<fileNames>
THE ANOMALY FORCING SIGNAL DATA NAME
</fileNames>
<offset>
0
</offset>
</fieldInfo>
```

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
