# Peer review of "Using the anomaly forcing Community Land Model (CLM 4.5) for crop yield projections"

_Geoscientific Model Development, 2020_

## Referee Comment (RC1) · Anonymous Referee #1 · 26 Oct 2020

This manuscript presented the first evaluation of the anomaly forcing mode for crop yield simulation with CLM4.5 in CESM. The authors created anomaly forcing datasets for three climate scenarios (1.5 °C warming, 2.0 °C warming, and RCP4.5) and conduct global CLM crop simulations using the compset of CLM45BGCCROP at a spatial resolution of 1.9 by 2.5 degrees. The authors found that the anomaly forcing CLM could not produce crop yields identical to the standard CLM with subdaily forcing, but captured the relative changes between scenarios and over time, as well as regional crop yield variations.

Overall, this manuscript is neat. It fits the "model evaluation" category of GMD and should be very interesting to the broader community. It is well written and organized. I only have the following minor concerns for the authors to consider.

[Figure]

it is not very clear to me how the authors calculated the "forcing variance R2" as shown in Fig. 1. The definition in the caption is unclear. Does "every ten year-averaged monthly variance" represent variance of very ten-year-averaged monthly forcing or I should interpret it by the wolds themselves? It would be good to also note the sample number for it, which would help the understanding.

I suggest the authors give more details on how to calculate the averaged yield across different crop species and regions for a specific country/region as shown in Fig. 4 and other maps. Is it simple area-weighted average?

L165: could you elaborate why the computational cost is high when using transient CO2 and nitrogen fertilization? Is the higher computation cost from the "transient CO2 and nitrogen fertilization" simulation itself (compared with constant CO2 and fertilization cases) or just more experiments?

L252-L253: what's the consideration for not masking the insignificant differences here?

In the discussion part, the authors discussed the potential causes for some exceptions, which is good. However, I suggest the authors give some example figures for those exceptional data, either in the main manuscript or in the supplementary materials. It would help strength the statements in this part.

Figure 4 is not referred in the manuscript at all.

L340-L341: "is due are due"->"are due"

It would be good to give some implications for CLM5.0 too in the final discussion part. For example, whether there is any changes of the anomaly forcing mode in CESM2.0 and whether the results for CLM4.5 still holds for CLM5.0. That would be also helpful.

---

## Referee Comment (RC2) · Anonymous Referee #2 · 2 Dec 2020

**General comments**

The authors demonstrate the impact of using anomaly forcing in the Community Land Model 4.5 on crop yield projections, as compared to using 3-hourly forcing data, for three scenarios: 1.5 °C warming, 2.0 °C warming, and RCP4.5. This is an important and timely piece of work, given that high resolution output data is not always easily available from climate models for use in driving crop components of land-surface models.

The paper is well written and includes all relevant information for reproducing the key results. I have a few specific comments below to be addressed before publication.

**Specific comments**

[Figure]

Line 28 "Our approach can be adopted by other land surface models to expand their capabilities for utilizing monthly climate data" Could you elaborate on this by adding a paragraph to the discussions section to discuss the applicability of this method and these results to other models?

Line 59: "biogeochemical compset is active" is jargon specific to CLM – could you replace with a more general phrase? (or add a sentence to explain what a "compset" is)

Line 59: could you indicate what "CLM-CN" and "CLM-BGC" include? (can be very brief e.g. what the "CN" and "BGC" stand for)

Line 74: add references for CRUNCEP, QIAN

Line 96: The phrase "has been in function" is not clear, so should be reworded. E.g. could replace with "has been functional" or "has been available".

Table 1: Define abbreviations CAM and MOAR.

Line 141: Change "multiplies" to "multiplied by"

Line 149: Change "equation2" to "equation 2"

Line 153-155: Need to explicitly define the quantities used in these equations. Also, are the underscores intentional, or should they be subscripts instead? Consider whether the notation for each variable could be simplified (e.g. is it necessary to include the letters "var", or is this implicit?).

Replace all occurrences of "CO2" with "$CO_2$" e.g. lines 164, 166.

Line 180: replace "R2" with "$R^2$".

Line 189: replace "as" with "to"

Line 194: explain "bottom atmosphere temperatures". Is this the air temperature of the lowest atmospheric level simulated by CESM? What height or pressure level is this?

Line 205-6: "we set the maximum precipitation anomaly ratio to 5 to avoid unrealistically extreme precipitation levels". Can you add an explanation of why is this necessary i.e. what are causing these extreme precipitation levels, with references.

Figure 2 caption: Change "1pt5, 2pt0" to "1.5 °C, 2.0 °C"

Line 237-9: "For irrigated crops, such overestimations in the northern US and Europe disappear (Figure 3g-i) because sufficient irrigation was added to the irrigated soil column; as long as there is plant water stress which removed water availability impacts on crop yields." Can you clarify this sentence, since at the moment it seems counter-intuitive (did you mean something like: "because sufficient irrigation was added to the irrigated soil column to prevent plant water stress, which removed water availability impacts on crop yields"?).

Fig 3 caption: "the historical crop map in 2005". Can you add the reference?

Lines 267-272: Is this the first time these "standard CLM" yield projections have been published? If yes, could you add a discussion, including a comparison to other yield projections in the literature for these scenarios. If not, could you add references.

Line 368: give reference for UN FAO yields

---

## Author Comment (AC1) · 10 Jan 2021

Dear Sir or Madam:

Thank for your time of reviewing our manuscript. We appreciate all your comments which largely improved the manuscript. The detailed replies are in blue. We hope these responses could fully address your comments.

Best wishes,

Yaqiong Lu and Xianyu Yang

**Anonymous Referee #2**

**General comments**

The authors demonstrate the impact of using anomaly forcing in the Community Land Model 4.5 on crop yield projections, as compared to using 3-hourly forcing data, for three scenarios: 1.5 °C warming, 2.0 °C warming, and RCP4.5. This is an important and timely piece of work, given that high resolution output data is not always easily available from climate models for use in driving crop components of land-surface mod- els.

The paper is well written and includes all relevant information for reproducing the key results. I have a few specific comments below to be addressed before publication.

**Specific comments**

Line 28 "Our approach can be adopted by other land surface models to expand their capabilities for utilizing monthly climate data" Could you elaborate on this by adding a paragraph to the discussions section to discuss the applicability of this method and these results to other models?

We added some discussions of the applicability of this method to other land surface models at line 385-388:

Our approach can be adopted by other land surface models to expand their capabilities for utilizing monthly climate data. The source code of the anomaly forcing CLM is available at post4.5crop_slevis/models/lnd/clm/src/cpl/ lnd_import_export.F90. The Fortran code could be transplanted to other land surface models which use NetCDF format atmospheric forcing.

Line 59: "biogeochemical compset is active" is jargon specific to CLM – could you replace with a more general phrase? (or add a sentence to explain what a "compset" is)

We use component which is easier for understanding.

Line 59: could you indicate what "CLM-CN" and "CLM-BGC" include? (can be very brief e.g. what the "CN" and "BGC" stand for)

We added descriptions at line 58-61:

"The crop model in CLM runs when the soil biogeochemical component is active, and it was tested with the CLM-CN in version 4.0 and tested with CLM-BGC in version 4.5, where CLM-CN and CLM-BGC are officially supported soil biogeochemical components in CLM4.0 and CLM4.5 respectively."

Line 74: add references for CRUNCEP, QIAN

We added references for CRUNCEP, QIAN at line 76:

"e.g., CRUNCEP (Viovy, 2018), QIAN (Qian et al., 2006)"

Viovy, N.: CRUNCEP Version 7 - Atmospheric Forcing Data for the Community Land Model. https://doi.org/10.5065/PZ8F-F017, Research Data Archive at the National Center for Atmospheric Research, Computational and Information Systems Laboratory, 2018.

Qian, T., Dai, A., Ternberth, K. E., and Olseon, K. W.: Simulation of Global Land Surface Conditions from 1948 to 2004. Part I: Forcing Data and Evaluations, Journal of Hydrometeorology, 7, 953-975, 2006.

Line 96: The phrase "has been in function" is not clear, so should be reworded. E.g. could replace with "has been functional" or "has been available".

We modified the phrase to has been available.

Table 1: Define abbreviations CAM and MOAR. Line 141: Change "multiplies" to "multiplied by" Line 149: Change "equation2" to "equation 2"

Here CAM is the Community Atmosphere Model, MOAR is the abbreviation of Mother Of All Runs. In the text, we modified CAM to Community Atmosphere Model and MOAR to the standard CLM forcing to avoid confusion. We also changed "multiplies" to "multiplied by" and "equation2" to "equation 2".

Line 153-155: Need to explicitly define the quantities used in these equations. Also, are the underscores intentional, or should they be subscripts instead? Consider whether the notation for each variable could be simplified (e.g. is it necessary to include the letters "var", or is this implicit?).

We simplified the terms and defined the quantities at line 155-159:

$$af_{i,j,m} = fut_{i,j,m} - hist_{i,j,m} \qquad (1)$$

$$af_{i,j,m} = fut_{i,j,m}/hist_{i,j,m} \quad (2)$$

Where $af_{i,j,m}$ is anomaly forcing signal at a location i and j in a month m, $fut_{i,j,m}$ is the averaged future value and $hist_{i,j,m}$ is the averaged historical value at a location i and j in a month m.

Replace all occurrences of "CO2" with "CO$_2$" e.g. lines 164, 166.

We replaced all CO2 with CO$_2$

Line 180: replace "R2" with "R$^2$".

Done.

Line 189: replace "as" with "to"

Done

Line 194: explain "bottom atmosphere temperatures". Is this the air temperature of the lowest atmospheric level simulated by CESM? What height or pressure level is this?

Yes, the bottom atmosphere temperature is the air temperature of the lowest atmospheric level. In our simulation, the bottom atmosphere temperature are simulated by CESM. CESM uses a hybrid terrain follow sigma coordinate at the bottom surface. The sigma vertical coordinate defined as the ratio of the pressure at a given point in the atmosphere to the pressure on the surface of the earth underneath it. The lowest sigma level in the CESM simulation we used is 0.9925. So the pressure of the lowest layer is 992.5 hPa if the surface pressure is 1000 hPa. The actual height of the lowest atmospheric level varies across gridcells.

Line 205-6: "we set the maximum precipitation anomaly ratio to 5 to avoid unrealistically extreme precipitation levels". Can you add an explanation of why is this necessary i.e. what are causing these extreme precipitation levels, with references.

Ratio 5 was suggested by NCAR scientists David Lawrence and Sean Swenson, who are core developers of CLM and wrote the initial anomaly forcing code in CLM.  Most of unrealistic extreme precipitation ratio are actually due to the nearly zero historical precipitation (the denominator). The cap for the precipitation anomaly ratio is use to avoid such situation.

Figure 2 caption: Change "1pt5, 2pt0" to "1.5 ˚C, 2.0 ˚C"

We modified the caption of Figure 2.

Line 237-9: "For irrigated crops, such overestimations in the northern US and Europe disappear (Figure 3g-i) because sufficient irrigation was added to the irrigated soil col- umn; as long as there is plant water stress which removed water availability impacts on crop yields." Can you clarify this sentence, since at the moment it seems counter- intuitive (did you mean something like: "because sufficient irrigation was added to the irrigated soil column to prevent plant water stress, which removed water availability impacts on crop yields"?).

For the rainfed crops, the anomaly forcing CLM had higher soil moisture at planting due to higher snow cover so the crop yield was higher in the anomaly forcing CLM. But for the irrigated crops, the standard CLM also received plenty of water from irrigation, so the water stress disappeard in standard CLM.

We clarified this sentence at line 241-243:

"For irrigated crops, such overestimations in the northern US and Europe disappear (Figure 3g-i) because sufficient irrigation was added to the irrigated soil column in the standard CLM, which removed the plant water stress that was seen for rainfed crops."

Fig 3 caption: "the historical crop map in 2005". Can you add the reference?

We added the url for the data in Figure 3 caption: MAPSPAM 2005; https://www.mapspam.info/

Lines 267-272: Is this the first time these "standard CLM" yield projections have been published? If yes, could you add a discussion, including a comparison to other yield projections in the literature for these scenarios. If not, could you add references.

The crop yield projections have been published in Ren et al., 2018. We added the citation at line 268: "in the standard CLM (Ren et al., 2018)"

Ren, X., Lu, Y., O'Neill, B. C., and Weitzel, M.: Economic and biophysical impacts on agriculture under 1.5 °C and 2 °C warming, Environ Res Lett, 13, 2018.

Line 368: give reference for UN FAO yields

We added the url for UNFAO crop yield statistics at line 372: http://www.fao.org/statistics/en/

---

## Author Comment (AC2) · 10 Jan 2021

Dear Sir or Madam:

Thank for your time of reviewing our manuscript. We appreciate all your comments which largely improved the manuscript. The detailed replies are in blue. We hope these responses could fully address your comments.

Best wishes,

Yaqiong Lu and Xianyu Yang

**Anonymous Referee #1**

This manuscript presented the first evaluation of the anomaly forcing mode for crop yield simulation with CLM4.5 in CESM. The authors created anomaly forcing datasets for three climate scenarios (1.5 $^{\circ}$C warming, 2.0 $^{\circ}$C warming, and RCP4.5) and con- duct global CLM crop simulations using the compset of CLM45BGCCROP at a spatial resolution of 1.9 by 2.5 degrees. The authors found that the anomaly forcing CLM could not produce crop yields identical to the standard CLM with subdaily forcing, but captured the relative changes between scenarios and over time, as well as regional crop yield variations.

Overall, this manuscript is neat. It fits the "model evaluation" category of GMD and should be very interesting to the broader community. It is well written and organized. I only have the following minor concerns for the authors to consider.

it is not very clear to me how the authors calculated the "forcing variance R2" as shown in Fig. 1. The definition in the caption is unclear. Does "every ten year-averaged monthly variance" represent variance of very ten-year-averaged monthly forcing or I should interpret it by the wolds themselves? It would be good to also note the sample number for it, which would help the understanding.

We added descriptions in the method section at line 202-203:

We calculated the variation for twelve months in each decade, so we have 7 decades and 12 months variance and the sample size is 84 when setting up the regression.

I suggest the authors give more details on how to calculate the averaged yield across different crop species and regions for a specific country/region as shown in Fig. 4 and other maps. Is it simple area-weighted average?

The integrated crop yield are area weighted crop yield. The crop area map we used was MAPSMAP (https://www.mapspam.info/) 2005 crop area. The regional average in Figure 4 are simply the regional average of integrated crop yield.

L165: could you elaborate why the computational cost is high when using transient CO2 and nitrogen fertilization? Is the higher computation cost from the "transient CO2 and nitrogen fertilization" simulation itself (compared with constant CO2 and fertilization cases) or just more experiments?

Using transient CO2 and nitrogen fertilization did not add extra computational cost. Here me mean computational cost due to more experiments.

L252-L253: what's the consideration for not masking the insignificant differences here?

We did not mask so the readers can have a better visualization on the detailed bias, even they are insignificance. Because I feel it would help some readers who cares about the overall bias.

In the discussion part, the authors discussed the potential causes for some exceptions, which is good. However, I suggest the authors give some example figures for those exceptional data, either in the main manuscript or in the supplementary materials. It would help strength the statements in this part.

We included three figures in the supplementary materials and referred these figures in our discussion. We hope that could strength the discussion. In particular, we add Figure S1 to show the grain fill days difference between anomaly forcing CLM and standard CLM; Figure S2 to show the percentage differences of leaf area index, gross primary production, soil water, latent heat flux, and sensible heat flux between anomaly forcing CLM and standard CLM; Figure S3 to show the percentage differences of boreal summer latent heat flux differences between anomaly forcing CLM and standard CLM.

[Figure]

Figure S1. 70-year averaged differences of grain fill days between the anomaly forcing CLM and the standard CLM for rice (a-c), tropical maize (d-f), and tropical soybean (g-i) for the 1.5ºC, 2.0 ºC, and RCP4.5 scenarios. All differences shown here are statistically significant differences tested by the Kolmogorov-Smirnov test with a sample size of 84. The gray areas are regions that did not show significant differences.

[Figure]

Figure S2. The percentage differences between the anomaly forcing CLM and the standard CLM for Leaf Area Index (LAI; a1-a3), Gross Primary Production (GPP; b1-b3), Soil Water (SW; c1-c3), Latent Heat Flux (LE; d1-d3), and Sensible Heat Flux (SH; e1-e3) for the 1.5ºC, 2.0 ºC, and RCP4.5 scenarios. All differences shown here are statistically significant differences tested by the Kolmogorov-Smirnov test with a sample size of 84. The gray areas are regions that did not show significant differences.

[Figure]

Figure S3. The percentage differences of boreal summer latent heat flux between the anomaly forcing CLM and the standard CLM for the 1.5°C, 2.0 °C, and RCP4.5 scenarios. All differences shown here are statistically significant differences tested by the Kolmogorov-Smirnov test with a sample size of 84. The gray areas are regions that did not show significant differences.

Figure 4 is not referred in the manuscript at all.

We referred figure 4 at line 269

L340-L341: "is due are due"->"are due"

We revised the typo at line 343.

It would be good to give some implications for CLM5.0 too in the final discussion part. For example, whether there is any changes of the anomaly forcing mode in CESM2.0 and whether the results for CLM4.5 still holds for CLM5.0. That would be also helpful.

We added some discussions of the implications for CLM5.0 at line 378-384:

The anomaly forcing method in CLM5.0 remains unchanged so the bias due to anomaly forcing may still exists in CLM5.0. For example, CLM5.0 uses the same threshold to differ rain and snow, so the bias due to higher snow cover in the Northern Hemisphere may still exists in CLM5.0. However, the crop model in CLM5.0 includes new features as reported in Lombardozzi et al., (2020). For example CLM5.0 uses time-varying spatial distributions of major crop types and has updated fertilization and irrigation schemes. These updates of crop model in CLM5.0 may improve crop  yields of anomaly forcing CLM5 compared to crop yield in reality.

---

## Author Response (AR2)

Dear Christoph,

Thank for the reminder, we read through our reply and make sure all the responses are showed in the text. We also revised the manuscript based on your other comments.

Sincerely,

Yaqiong and Xianyu

**Topical Editor Decision: Publish subject to minor revisions (review by editor)** (14 Jan 2021)
by Christoph Müller
Comments to the Author:
Dear Dr. Lu and Dr. Yang, thank you for revising the manuscript in accordance with the reviewers' advise.
Some of the comments have, however, only been answered in the response letter but did not lead to any changes in the manuscript so that readers may have similar questions as the reviewers but do not see the answers. Please also modify the manuscript to provide the clarifications provided in the response letter. Examples include:

* reviewer1, area weighting in aggregates

We added the text to line 187-190 (the line number is based on the track change version which is attached with this response):
"For the crop yield analysis, we aggregated the individual crop yield into an integrated crop yield by area weighted mean based on the crop area map MAPSMAP (https://www.mapspam.info/) 2005 crop area. The regional crop yield was simply the regional averaged crop yield at 9 regions defined in Ren et al., (2018).
"

* reviewer1, computational costs
We added the text to line 182-185:
"The transient $CO_2$ and nitrogen fertilization did not add extra computational cost compared to the constant $CO_2$ and nitrogen fertilization simulation. However, due to our limited computational resources could not afford more experiments, we only tested such responses for the RCP4.5 scenario."

* reviewer1, masking of insignificant areas in Figure 2d-i
We clarified in the figure caption at line 288: "For plots d-i, we did not mask the insignificant differences to show an overall bias."

* reviewer2, bottom atmosphere layer

We clarified it at line 220: "bottom layer atmosphere temperatures (sigma vertical coordinate, σ=0.9925)"

* reviewer2, ratio 5
We added text to line 236-240:
"Ratio 5 was suggested by NCAR scientists David Lawrence and Sean Swenson, who are core developers of CLM and wrote the initial anomaly forcing code in CLM. Most of unrealistic extreme precipitation ratio are actually due to the nearly zero historical precipitation (the denominator of equation 2). The cap for the precipitation anomaly ratio is use to avoid such situation."

I disagree with the speculation on how the annomaly forcing will work in CLM5.0. New model features, such as fertilization schemes may improve the general ability of the model to reproduce observed crop yield levels, but that also was not reported as a problem in the standard forcing setup. I see that reviewer1 asked for some speculation on how this would perform with CLM5.0, but I would recommend to clearly mark this as speculation and suggest that the annomaly forcing will have to be tested for suitability in any new version of CLM and for the specific research question under investigation (e.g. if the interest is in absolute yield levels or in qualitative differences).

Thanks for the comment. We actually mean the same thing, where the improved crop model in CLM5 could improve crop yield simulation for both standard CLM and anomaly forcing CLM. We agree that testing for the suitability of new model is necessary. We revised our discussion at line 416-428:

"The crop model in the most recent version of CLM5.0 includes new features as reported in Lombardozzi et al., (2020). For example CLM5.0 uses time-varying spatial distributions of major crop types and has updated fertilization and irrigation schemes. These updates of crop model in CLM5.0 may improve the crop yield simulations for both standard CLM and anomaly forcing CLM compared to crop yield in reality. The anomaly forcing method in CLM5.0 remains unchanged so we speculate the bias due to anomaly forcing may still exist in CLM5.0. For example, CLM5.0 uses the same threshold to differ rain and snow, so the bias due to higher snow cover in the Northern Hemisphere may still exists in CLM5.0. However, how will the magnitude of the bias change is unclear. We suggest that the anomaly forcing of CLM5.0 to be tested if the research interest is in absolute yield or in qualitative difference."

The new Figure S1 does not provide the unit of changes shown. All other figures mention the unit in the figure caption, but I would generally recommend to include units with the legend bar within the figures, as these may be used in presentations or similar and missing units can lead to confusion.

We added the unit to legend bars.

You provide a path to the anomaly forcing source code, but don't explicitly say that this (presumably?) refers to the code given in the zenodo archive. Please add information on where that path can be found.

We added more detail at line 431-434:
"The source code of the anomaly forcing CLM is available at the repository website Zenodo https://doi.org/10.5281/zenodo.3900671. The path is post4.5crop_slevis/models/lnd/clm/src/cpl/ lnd_import_export.F90 when unzip post4.5crop_slevis_codeforGMD.tar.gz."

I'm happy to accept the manuscript as soon as these points have been addressed.

Best regards
Christoph

[revised manuscript text omitted]
 Leaf Area Index (LAI; a1-a3), Gross Primary Production (GPP; b1-b3), Soil Water (SW; c1-c3), Latent Heat Flux (LE; d1-d3), and Sensible Heat Flux (SH; e1-e3) for the 1.5°C, 2.0 °C, and RCP4.5 scenarios. All differences shown here are statistically significant differences tested by the Kolmogorov-Smirnov test with a sample size of 84. The gray areas are regions that did not show significant differences.

[Figure]

Figure S3. The percentage differences of boreal summer latent heat flux between the anomaly forcing CLM and the standard CLM for the 1.5ºC, 2.0 ºC, and RCP4.5 scenarios. All differences shown here are statistically significant differences tested by the Kolmogorov-Smirnov test with a sample size of 84. The gray areas are regions that did not show significant differences.